

# A critical trade-off between nitrogen quota and growth allows *Coccolithus braarudii* life cycle phases' to exploit varying environment

Joost de Vries[1,2], Fanny Monteiro[1], Gerald Langer[3], Colin Brownlee[2], and Glen Wheeler[2]

[1]BRIDGE, School of Geographical Sciences, University of Bristol, University Road, Bristol BS8 1SS, UK
[2]The Marine Biological Association of the United Kingdom, The Laboratory, Citadel Hill, Plymouth PL1 2PB, UK
[3]Institute of Environmental Science and Technology (ICTA), Universitat Autonoma de Barcelona, 08193 Barcelona, Spain

**Correspondence:** Joost de Vries (joost.devries@bristol.ac.uk)

**Abstract.** Coccolithophores have a distinct haplo-diplontic life cycle, which allows them to grow and divide in two different life cycle phases (haploid and diploid). These life cycle phases vary significantly in inorganic carbon content and morphology, and inhabit distinct niches, with haploids generally preferring low-nutrient and high-temperature and -light environments. This niche contrast indicates different physiology of the life cycle phases, which is considered here in the context of a trait trade-off

framework, in which a particular set of traits comes with both costs and benefits. However, coccolithophore's phase trade-offs are not fully identified, limiting our understanding of the functionality of the coccolithophore life cycle. Here, we investigate the response of the two life cycle phases of the coccolithophore *Coccolithus braarudii* to key environmental drivers: light, temperature and nutrients, using laboratory experiments. With this data, we identify the main trade-offs of each life cycle phase and use models to test the role of such trade-offs under different environmental conditions.

The lab experiments show the life cycle phases have similar cell size, nitrogen requirement, uptake rates, and temperature and light optima. However, we find that they have different coccosphere sizes, maximum growth rates and nitrogen quotas. We also observe a trade-off between maximum growth rate and nitrogen quota, with higher growth rates and small nitrogen storage in the haploid phase and vice versa in the diploid phase.

Testing these phase characteristics in the model, we find that the growth-quota trade-off allows *C. braarudii* to exploit

variable nitrogen conditions more efficiently. Because while the diploid ability to store more nitrogen is advantageous when the nitrogen supply is intermittent, the higher haploid growth rate is advantageous when the nitrogen supply is constant.

Although the ecological drivers of *C. braarudii* life cycle fitness are likely multi-faceted, spanning both top-down and bottom-up trait trade-offs, our results suggest that a trade-off between nitrogen storage and maximum growth rate is an essential bottom-up control on the distribution of *C. braarudii* life cycle phases.

# 1   Introduction

Coccolithophores are important contributors to the carbon cycle. Their influence stems from their impact on the organic carbon pump through photosynthesis and the inorganic carbon pump through calcium carbonate production (Zeebe, 2012; Passow and





Carlson, 2012; Boyd et al., 2019). Although both carbon pumps influence the global ocean carbon cycle, they have opposing effects. While the organic carbon pump sequesters carbon, the inorganic carbon pump releases carbon back into the atmosphere

on short time scales (less than $10^4$ years) (Zeebe, 2012).

Coccolithophores utilize a haplo-diplontic life cycle, which makes them distinct from other phytoplankton and allows them to grow and divide in both the haploid and diploid life cycle phases (Dassow and Montresor, 2011; Frada et al., 2018). These two life cycle phases have different inorganic and organic carbon contents, with the diploid life cycle phase generally more heavily calcified (Cros et al., 2000; Young et al., 2003; Daniels et al., 2016; Fiorini et al., 2011a, b; Frada et al., 2018). The two life

cycle phases furthermore tend to have distinct niches, with the haploid cells found in lower nutrients and higher temperature and light environments (de Vries et al., 2021). These distinct niches and differences in PIC:POC ratios imply coccolithophore life cycle phases impact the organic and inorganic carbon pump differently. Nonetheless, the differential impact of coccolithophore life cycle phases is poorly understood, partly because we lack a quantitative understanding of the drivers behind the haploid and diploid distribution.

Physiology drives the growth response of phytoplankton to the environment. Understanding the physiological differences between coccolithophore life cycle phases and the subsequent impact on growth rate is thus key to understanding coccolithophore life cycle distribution. Here we investigate the physiological differences of the coccolithophore *Coccolithus braarudii* within a trait-based framework to contextualize our results. In this framework, each trait is considered to have both costs and benefits, thus presenting trade-offs for organisms utilizing such traits (Kiørboe et al., 2018). Since trait-based approaches focus on traits

rather than individual species, these results can be applied to other coccolithophore species.

*Coccolithus braarudii* is a key coccolithophore species in the Arctic Ocean, where it dominates coccolithophore calcium carbonate production because of its much larger size and calcium carbonate content than *Emiliania huxleyi*, which numerically dominates this region (Daniels et al., 2014). The life cycle phases of *Coccolithus braarudii* are morphologically distinct, with the haploid life cycle phase utilizing a holococcolith (HOL) morphology and the diploid life cycle phase utilizing a

heterococcolith (HET) morphology (Houdan et al., 2004). Most lab work on *C. braarudii* has focused on the HET phase, with research on the HOL life cycle phase much more limited (Houdan et al., 2006; Langer et al., 2022).

The distribution of *Coccolithus braarudii* varies in the ocean with *C. braarudii* HET dominating high-nutrient and turbulent regions, while *C. braarudii* HOL is primarily found in lower-nutrient and more stratified regions (Malinverno et al., 2009; D'Amario et al., 2017). While some previous work suggests that differential response to turbulence (Houdan et al., 2004) and

light (Langer et al., 2022) could drive this distribution difference, open questions remain about how physiology differs between the two life cycle phases.

For our experiments on *C. braarudii* we focus on cell size, coccosphere size, nutrient uptake rates, nutrient quotas, and photosynthetic response (Fig. 1). These traits have been shown to influence phytoplankton distribution and can be defined in deterministic modelling frameworks (Hansen, 1994; Hansen et al., 1997; Ward et al., 2017; Litchman et al., 2007; Follows

et al., 2007; Dutkiewicz et al., 2015). For instance, cell size influences nutrient uptake rates (Ward et al., 2017; Litchman et al., 2007), while coccosphere size influences grazing dynamics by increasing the coccolithophore diameter (Hansen, 1994; Hansen et al., 1997). Nutrient quota influences nutrient requirement (Litchman et al., 2007) and storage (Falkowski and Oliver,



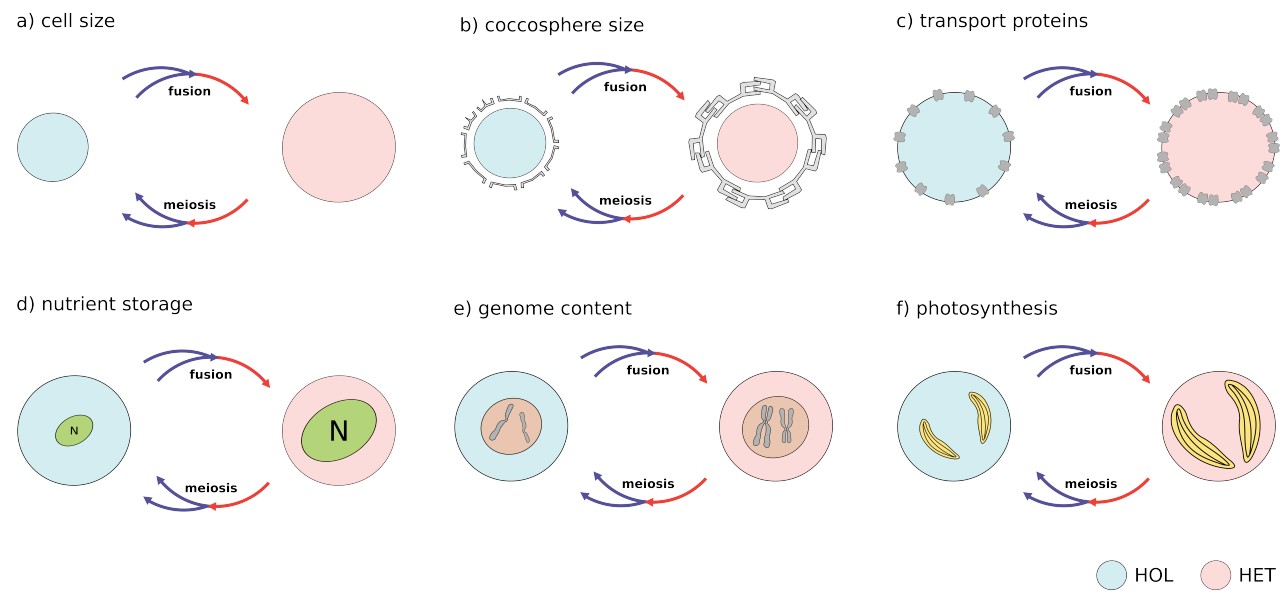

**Figure 1.** Main coccolithophore traits investigated in this study: **(a)** cell size; **(b)** coccosphere size; **(c)** transport proteins; **(d)** nutrient storage; **(e)** genome content; **(f)** photosynthesis.

2007; Grover, 1991). Photosynthetic response influences light optima and has previously been suggested to be a key driver in *Coccolithus braarudii* life cycle distribution (Langer et al., 2022). We also investigated differences in temperature optima, as
this is a major driver of coccolithophore life cycle phases in situ (de Vries et al., 2021).

We compare three *C. braarudii* HET and three *C. braarudii* HOL life cycle phases for response to light, nutrients, and temperature. In addition, we measure their nitrogen and DNA contents. Finally, we contextualise our results with a mechanistic model to investigate whether the trait differences provide any competitive advantage to each of the life cycle phases.

This research elucidates what fundamentally separates the two life cycle phases from a bottom-up perspective and lays the
groundwork for incorporating the coccolithophore life cycle into ecosystem models.

## 2 Methods

### 2.1 Experimental conditions

The *Coccolithus braarudii* strains utilised in our study are the HET strains PLY182g, RCC6535, RCC1200, and the HOL strains RCC1203, RCC3777 and RCC3779. PLY182g was obtained from the Marine Biological Association (MBA) culture
collection, while we obtained the rest of the cultures from the Roscoff Culture Collection (RCC). All strains were maintained

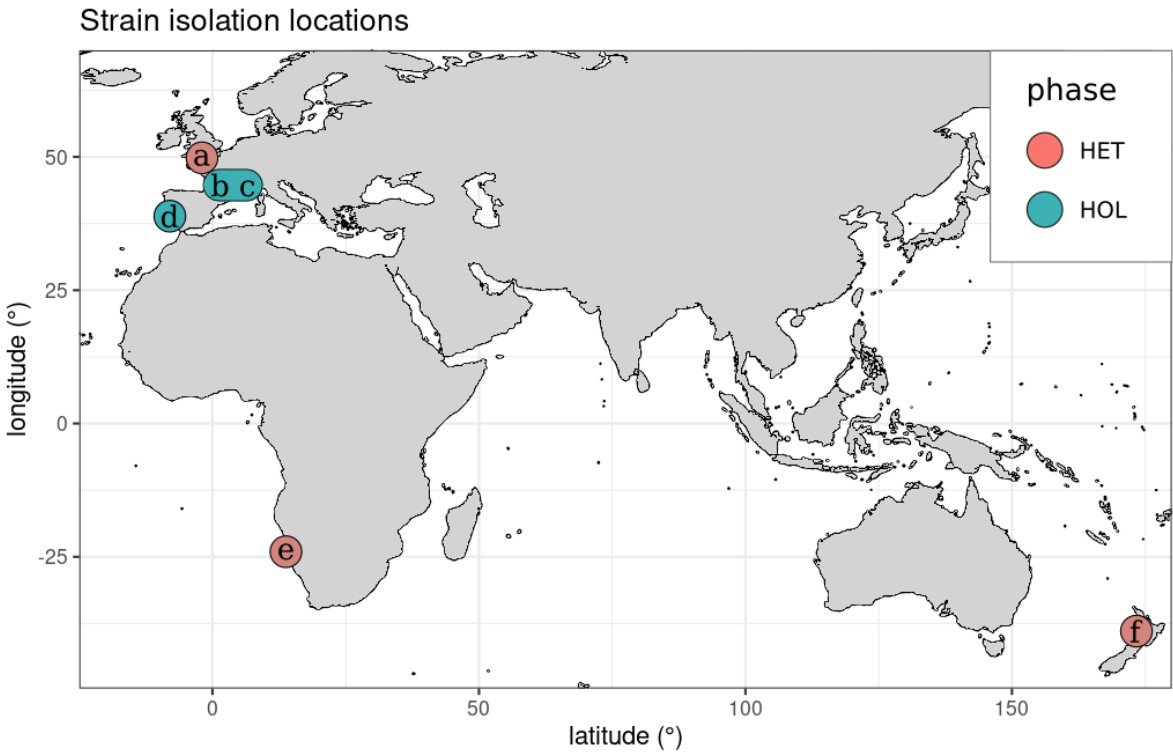

**Figure 2.** Isolation locations of the strains utilised in this study. The six strains investigated were: **(a)** PLY182g (HET); **(b)** RCC1203 (HOL); **(c)** RCC3777 (HOL); **(d)** RCC3779 (HOL); **(e)** RCC1200 (HET); **(f)** RCC6535 (HET).

using K/2 media (Probert and Houdan, 2004), which we prepared using water from the L1 station in the Plymouth sound (Smyth et al., 2010). All cultures were maintained at 15°C in a temperature-controlled room, at 50 $\mu$E m$^{-2}$ s$^{-1}$ under a 12:12 L:D cycle.

Most strains were isolated near coastal regions in Europe (Fig. 2), except RCC1200, which was isolated off the coast of Nambia and RCC6535, which was isolated in New Zealand. All HOL strains were collected during the winter, while the HET strains RC6535 and RC1200 were collected in the summer. The collection date of PLY182g is unknown, but as this species tends to only occur in Plymouth during the winter (Widdicombe et al., 2010), it was likely collected during this period.

## 2.2 Nutrient limitation

Nutrient limitation experiments were conducted by modifying the K/2 media, which contained 220.5 $\mu$M by reducing $NO_3$ to 20 $\mu$M. At this nitrogen concentration, final cell counts reached approximately 20,000 cells ml$^{-1}$, a cell density high enough for carbon and nitrogen analysis but low enough that we did not observe the effects of unbalanced carbonate chemistry (Langer et al., 2022).





## 2.3 Cell size

We measured cell size by imaging cells using a Leica DMi8 inverted microscope and analysed them using the Python package
scikit-image (van der Walt et al., 2014).

Specifically, we segmented the images using scikit-image sobel segmentation (Kroon, 2009), and then manually sorted
regions of interest by removing out-of-focus images, clusters and detritus. Assuming spherical cells, we estimated cell volume
based on the mean of the longest diameters of each image.

Cell size was measured for all strains and both repleted and depleted cultures. Microscopy images were taken at the same
time of day (11 am) to minimise the influence of cell divisions on cell size across strains (Kottmeier et al., 2020).

Before imaging, the HET and HOL strains were decalcified by exposing the cultures to 1M HCL to decrease the pH to 3 and
then adding 1M NaOH to restore the pH. HOL cultures were treated for 1 minute and HET coccolithophores for 5 minutes as
we found these to be the shortest durations that decalcify the cells.

## 2.4 Growth rates

For replete experiments, growth rates were estimated using a BMG clario fluorescence plate using Chlorophyll fluorescence
excitation at 420 nm and at 680 nm - which were calibrated against manual Sedgewick Rafter Counter (SRC) counts. For
nutrient-replete cultures, this method worked well, with $R^2$ values above 0.95 for all calibrations. Growth rates for nutrient-
deplete experiments were estimated using only SRC counts, as fluorescence was found to be a poor proxy ($R^2 < 0.8$), likely
due to an influence of nutrient depletion on chlorophyll concentrations and subsequently fluorescence.

For each experiment, cultures were inoculated at approximately 500 cells ml$^{-1}$ and grown until approximately 10-20k cells
ml$^{-1}$. This initial concentration was chosen as this is near the detection limit of the fluorescence plate reader.

Cell density was measured daily. Growth rate was then estimated by fitting a linear model to log-transformed counts, includ-
ing only points exhibiting exponential growth and removing counts in the stationary phase.

## 2.5 Temperature optima

To measure the temperature response of the different strains. Each strain was grown in triplicate under nutrient-replete condi-
tions under 50 $\mu$E m$^{-2}$ s$^{-1}$ of light at 14, 15, 18, and 20°C and a 12:12 L:D light cycle. Before performing the temperature
growth rate experiments, the cultures were acclimated for 5 generations. The growth rate for each temperature was measured
using fluorescence with the BMG plate reader by inoculating cultures at 500 cells ml$^{-1}$ and then measuring fluorescence daily
until the cellular concentration of approximately 20,000 cells ml$^{-1}$ was reached.

## 2.6 Light optima

For the light experiment, we tested seven light intensities (20, 50, 70, 100, 110, 140 $\mu$E m$^{-2}$ s$^{-1}$). All light sensitivity ex-
periments were conducted at 18°C as this was found to be the optimal temperature for all strains in our temperature-sensitive
experiments. Prior to the light sensitivity experiments, the cultures were acclimated for 5 generations. These acclimated cul-



tures were then used to create new fluorescence-to-count calibration curves for the growth rate experiments. For the experiment,
a full spectrum Kessil A360W LED with 380 and 400 nm UV bands was used, which was kept on its whitest setting and a
12:12 L:D cycle. PAR for each treatment was determined using an LI-250A Photometer (LI-COR).

**Nutrient limitation and quotas**

We estimated the minimum and maximum nutrient quotas, called $Q^{min}$ and $Q^{max}$, for all five strains. For $Q^{max}$, the maximum
nitrogen content of a cell, nitrogen analysis of replete cells under exponential growth was used. This analysis was conducted
by filtering the samples onto glass GF/F filters, freeze drying, and then combusting the samples in a CN elemental analyser
which was conducted by OEA Laboratories Limited (Exeter, UK).

The minimum nutrient quota ($Q^{min}$), was estimated by dividing the absolute nutrient concentration at the start of the exper-
iment by the number of cells present once the cultures reached the stationary phase as described in Perrin et al. (2016).

### 2.6.1 Response of photosynthetic efficiency to nutrient limitation

Photosynthetic efficiency (Fv/Fm) and electron transport rates (ETR) were measured using a Walz water PAM for both repleted
and depleted cultures. For the nitrogen deplete experiment cultures were grown under 220 uM $NO_3$ and PAM was measured
during the exponential growth phase. For the deplete experiments, cell growth was measured daily using cell counts, and
PAM was conducted once cells ceased to divide. Before the PAM measurements, cells were dark acclimated for 1 hour under
temperature-controlled conditions. For the measurements, Fv/Fm was taken before measuring ETR response to PAR.

### 130 2.6.2 DNA extraction

DNA content for each strain was measured following methods described by Liefer et al. (2019). In brief, the cultures were
grown under nutrient-replete conditions in triplicate and harvested around 20k cells ml$^{-1}$. The DNA was then extracted using
n-lauryl sarcosine as the surfactant in the extraction protocol. It was then stained with SYBR Green II (Thermofisher S7564)
and quantified using a fluorescence-based microplate analysis at 490 nm, which was calibrated against known DNA content
stocks (Sigma, D4522-1MG). DNA was measured only for repleted cultures, and for depleted cultures, it was assumed to be
half that of replete cells - based on the assumption that cell division had ceased in such cultures.

### 2.7 Statistical analysis

To determine if HOL and HET cell size, coccosphere size, growth rates, DNA content, nutrient quotas, and nutrient uptake
rates were significantly different, we conducted a two-sample t-test for each parameter.
The influence of the life cycle phase on temperature and light optima was determined by fitting a generalized additive model
(GAM), which included the life cycle phase as a predictor. Where, if the phase was found not to have a significant influence on
model fit (p<0.05), the environmental response was interpreted as not being life cycle phase specific.





Analysis was conducted in R, and the t-test was conducted with the "stats" package (R Core Team, 2022), while the GAM analysis was done with the "mgcv" package (Wood, 2011). Before fitting each GAM, growth data was Min-Max normalized
for each species individually to remove the influence of ploidy-specific maximum growth rate differences.

## 2.8   Modelling

Nutrient uptake experiments are tedious and expensive using reactor experiments. We thus follow an alternative strategy as proposed by Perrin et al. (2016) fitting models to batch culture experiments to derive the parameters.

Our modelling strategy in this manuscript is thus twofold: first, we fit a model to experimental data to derive nutrient uptake
parameters. We then use this data for a second model, simulating resource competition of the two life cycle phases under different nutrient conditions and physiological parameterisations (so-called 'sensitivity experiments').

### 2.8.1   Model for nutrient uptake estimations

Mechanistic modelling relies on our understanding of cellular physiology to represent cell growth and nutrient limitation processes. This can be done with varying degrees of complexity ranging from simple models such as the Monod model,
which assumes a direct relationship between nutrient concentrations and growth rates (Monod, 1949), to very complex models such as that by Inomura et al. (2020), which represent individual cellular processes such as nutrient storage, transfer and reorganisation of cellular machinery. Here we utilise an internal quota cellular model, an intermediate complexity mechanistic model, which accounts for cellular nutrient storage but does not trace individual cellular processes. Such models can take several forms, including Droop and linear models as described in detail in Flynn (2008). Here we employ the normalised
rectangular hyperbolic (RH) model as proposed by Flynn (2008), which includes a response variable, which modulates the relationship between nutrient quota and growth.

Conceptually, the RH internal nutrient model regulates growth rates based on the available internal nutrients, which depends on the nutrient uptake rate and the concentration of external nutrients (Eq.1-6). The model traces environmental nutrient concentration ($N$), the cellular concentration ($P$), and the internal nutrient quota ($Q$) using the following three differentials:

$$\frac{dP}{dt} = \mu \cdot P \tag{1}$$

$$\frac{dN}{dt} = -V \cdot P \tag{2}$$

$$\frac{dQ_N}{dt} = V - \mu \cdot Q \tag{3}$$

where $\mu$ is the growth rate (d$^{-1}$) and $V$ is the nutrient uptake rate ($\mu$mol cell$^{-1}$ d$^{-1}$).

The modelled growth rate $\mu$ depends on the nutrient quota, which we define here following Flynn (2008)'s RH formulation:





$$\mu = \mu^{max} \cdot \frac{(1+KQ)(Q-Q^{min})}{(Q-Q^{min})+KQ(Q^{max}-Q^{min})} \tag{4}$$

where $\mu_{max}$ (d$^{-1}$) is the maximum growth rate under nutrient-replete conditions, $Q^{min}$ (pgN cell$^{-1}$) is the minimum cellular nutrient quota, $Q^{max}$ (pgN cell$^{-1}$) is the maximum cellular nutrient quota, and $KQ$ is a dimensionless parameter that determines the relationship between the internal nutrient quota and growth rate.

We estimate $KQ$ based on $Q^{min}$ and $Q^{max}$ as defined by Flynn (2008):

$$KQ = \frac{Q^{min}}{Q^{max}-Q^{min}} \tag{5}$$

In this equation, strains with a small ratio of minimum to maximum nutrient quotas have a strong linear relationship between growth rate and nutrient quota. While strains with a large minimum to maximum nutrient quota ratio exhibit a non-linear relationship between nutrient quota and growth (Flynn, 2008).

The cellular nutrient quota ($Q$) is replenished based on the cellular nutrient uptake rate, which we define here using Michaelis-Menden uptake kinetics, while also limiting uptake when the cellular nutrient quota is full:

$$V = V^{max} \cdot \frac{[N]}{[N]+KN} \cdot \frac{Q_i^{max}-Q_i}{Q_i^{max}-Q_i^{min}} \tag{6}$$

where, $V_{max}$ is the per cellular nutrient uptake rate (in $\mu$mol$_R$ cell$^{-1}$ d$^{-1}$) and $K_N$ is the Michaelis-Menten half-saturation constant (in $\mu$mol l$^{-1}$).

### 2.8.2 Estimating nutrient uptake rate using the internal storage model

The internal stores model used in our study includes six key parameters (see Eq. 1-6): $\mu^{max}$, $V^{max}$, $Q^{min}$, $Q^{max}$, $KN$ and $KQ$ (Table 1).

We empirically measured and estimated Q$^{min}$ and Q$^{max}$ from our lab experiments (see Section 2.6). The other parameters were estimated through previously described relationships or fitting models to our lab data, as this approach is cost and time-effective and works well (Perrin et al., 2016). We calculated $KQ$ using Eq. 5 and approximated $\mu^{max}$, $V^{max}$ and $KN$ using the models described below.

### 2.8.3 Maximum growth rate

We estimated the maximum growth rate ($\mu^{max}$) by fitting the sigmoidal function described by Zwietering et al. (1990) and Ward et al. (2017) to our nutrient limitation experiment data.

$$y = \frac{A}{1+\exp[(4\mu^{max}/A)(\lambda-t)+2]} \tag{7}$$





where A is the log of the maximum cell density (cells ml$^{-1}$) observed in the experiment, $\lambda$ is the lag time (d$^{-1}$) defined as the time the tangent to the curve at its inflection point equals 0.

### 2.8.4   Maximum nutrient uptake rate

We estimated the maximum nutrient uptake rate ($V^{max}$) for each strain by simplifying the equation proposed in Verdy et al. (2009):

$$\mu^{max} = \frac{\mu^{\infty} V^{max}(Q^{max} - Q^{min})}{V^{max} Q^{max} + \mu^{\infty} Q^{min}(Q^{max} - Q^{min})} \tag{8}$$

Assuming that $\mu^{max} \approx \mu^{\infty}$ as proposed by Ward et al. (2017) for cells $>10\mu$m and re-arranging Eq. 8 for $V^{max}$ lead to the equation:

$$V^{max} = \frac{\mu^{max} Q^{min}}{1 + Q^{min}} \tag{9}$$

### 2.8.5   Maximum nutrient uptake rate

For the remaining unknown parameter, the half-saturation constant for nutrient uptake ($KN$), we used a non-linear least squares fitting procedure to fit the internal nutrient stores model to our laboratory data. This fitting was conducted by minimising error through 'Basin-Hopping' of mean squared errors calculated for each set of parameter estimations (Wales and Doye, 1997). Basin-Hopping was conducted with the 'basinhopping' function in Python from the scipy.optimise library (Virtanen et al., 2020).

We constrained $KN$ to be positive as empirically, the half-saturation constant for growth cannot be negative. This was done by forcing the mean squared error value to infinity if bounds were not met (a high mean squared error value indicates a bad fit). Bounds for $KN$ were defined the same for all strains and set to be between $1e^{-99}$ and $1e^{99}$.

Initial fitting $KN$ estimates were based on *E. huxleyi* data from Perrin et al. (2016) and set to values of 0.35 $\mu$mol l$^{-1}$.

### 2.8.6   Chemostat model

To better visualise and interpret the impact of the different nutrient dynamics of the different coccolithophore strains, we utilise a 'chemostat' model to simulate competition.

To better visualise and interpret the impact of the different nutrient dynamics of the different coccolithophore strains, we developed a numerical 'chemostat' model. This model simulates resource competition between the HET and HOL strains, differentiated by their maximum growth rates and nutrient storage, as observed in the lab experiments.

For our chemostat model, we forced fluxes of nitrogen concentration using a pulse wave, which we define using a Fourier series expansion:



$$N(t) = A\frac{\tau}{T} + \frac{2A}{\pi} \sum_{n=1}^{\infty} \left( \frac{1}{n} \sin\left(\pi n \frac{\tau}{T}\right) \cos\left(2\pi n \frac{1}{T} t\right) \right) \tag{10}$$

where A is the pulse amplitude ($\mu$M), $\tau$ is the pulse length (d$^{-1}$), $T$ is the pulse period (d$^{-1}$), t is time (d), and n (unitless) is the integer multiple (which determines the "squareness" and is set as 100).

To model phytoplankton abundance in our model, we follow the equations described in Follows et al. (2018). However, we imposed a quadratic mortality term to represent viral lysis, grazing pressure and sinking as this is proposed to be more realistic (Steele and Henderson, 1992). We also included a chemostat turn-over term $\kappa$ (d$^{-1}$):

$$\frac{dP_i}{dt} = \mu_i P_i - \kappa P_i - m P_i^2 \tag{11}$$

The growth rate ($\mu_i$) in this model follows the Flynn (2008) internal nutrient model as described in Eq. 4.

Following Follows et al. (2018), we modelled the internal nutrient quota as a function of nutrient uptake rate and growth with a quadratic mortality term:

$$\frac{dQ_i}{dt} = V_i - \mu_i Q_i - m Q_i^2 \tag{12}$$

where $V_i$ is limited as the quota approaches the maximum value to prevent excess nitrogen accumulation following Eq. 6.

For all our numerical experiments, we used the Python Scipy 'solve_ivp' function to integrate the system of ordinary differential equations using the RK45 algorithm described by Dormand and Prince (1980).

### 2.8.7 Sensitivity experiment

We explored the sensitivity of the maximum nutrient quota ($Q^{max}$), as this is the main difference observed in our lab results.

Mechanistically, a high $Q^{max}$ relates directly to nutrient storage and, in theory, benefits the organism when nutrient supply is intermittent. Following the trait-based theory, $Q^{max}$ should also have a cost; otherwise, it would lead to the evolution of 'super organisms' out-competing everything else (Follows and Dutkiewicz, 2011). Among the potential costs of nutrient storage, the metabolic cost seems the most compelling cost since $Q^{max}$ it is inversely related to maximum growth in our lab results, and as specified by previous literature (Grover, 1991).

We explored the competitive advantage of a higher $Q^{max}$ under different nutrient intermittence input regimes, testing nutrient inputs ranging from steady states to inputs only once every four days. We did this by comparing relative HET and HOL abundances under different input scenarios with and without a metabolic cost of nutrient storage (Table A1).





## 3   Results

### 3.1   Cell and coccosphere size

In our lab cultures, all *C. braarudii* strains have a similar cell diameter (with a mean diameter of $18.2 \pm 0.8$ $\mu$m) (Fig. 3a) with no significant difference between HOL and HET strains (p-value = 0.195). However, a significantly larger coccosphere size is observed for the HET strains ($23.8 \pm 2.4$ $\mu$m) compared to the HOL strains ($19.5 \pm 2.2$ $\mu$m, p-value $< 0.001$) (Fig. 3b).

The similarity in cell diameter but not in coccosphere size between the two life cycle phases suggests that the phases have the same size trait trade-off from a nutrient perspective but a different trait trade-off regarding grazing susceptibility. The coccosphere volume does not impact nutrient uptake dynamics on the first order since it does not increase the cell's surface area but may impact grazing dynamics.

### 3.2   Light and temperature optima

Our lab experiments showed that all strains have a temperature optimum at 18°C with a steep decline in growth at higher and lower temperatures (Fig. 4a). There is no significant difference in temperature optima between the two life cycle cycle phases, as shown in our generalized additive model (p-value = 0.061).

Unlike temperature, response to light was found to be strain-specific (Fig. 4b), with no significant differences between the different life cycle phases (p-value = 0.383). For example, only HET PLY182g and HOL RCC1203 display growth inhibition at 150 $\mu$E m$^{-2}$ s$^{-1}$, while the other strains do not. We note that our tested irradiance values are much lower than potentially experienced in situ, which can reach over 1,000 $\mu$E m$^{-2}$ s$^{-1}$ in surface waters (Laliberté et al., 2016).

Compared to the work by (Langer et al., 2022), we notice stronger growth inhibition from 150 $\mu$E m$^{-2}$ s$^{-1}$ for the strain PLY182g. This difference could be due to experimental temperature differences, with our experiment conducted at 18°C instead of 15°C. The difference could also be due to the difference in light sources used in the two experiments, as we used an LED light with a UV band (390 to 400 nm) instead of the fluorescent light source with no UV used in Langer et al. (2022).

Nonetheless, the trends in our study for the two strains are similar to those reported by Langer et al. (2022), with PLY182g HET showing a more potent light inhibition than RCC3777 HOL. Here, by analyzing a series of strains, we find that this difference is strain- and not ploidy-specific, as both HET and HOL life cycle phases exhibit differences in response to irradiance.

### 3.3   Maximum specific growth rates

Our light and temperature optima experiments suggest that overall HOL maximum growth rates ($0.73 \pm 0.07$ d$^{-1}$) are significantly higher than HET maximum growth rates ($0.56 \pm 0.07$ d$^{-1}$, p-value $< 0.001$; Fig. 5). This agrees with Langer et al. (2022), who reports maximum growth rates of 0.5 and 0.9 for PLY182g HET and RCC3777 HOL, respectively.



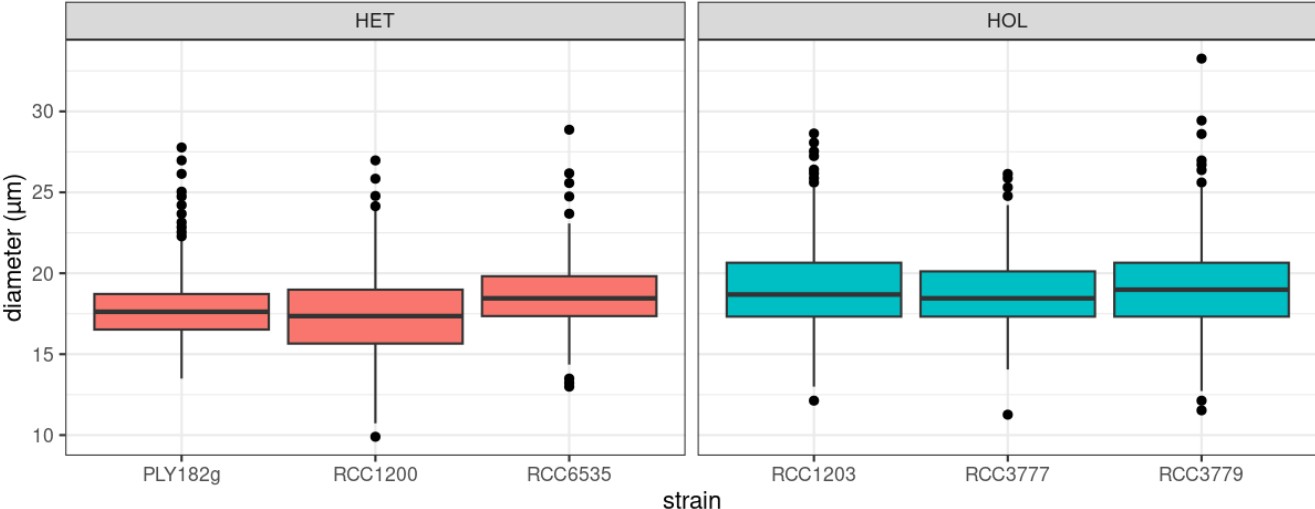

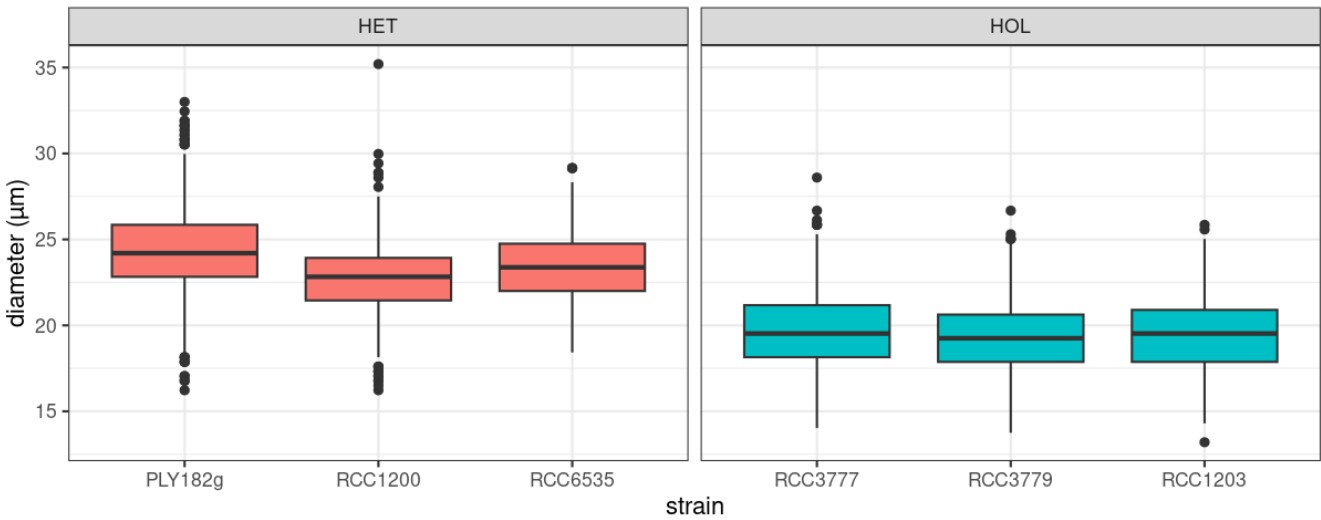

**Figure 3.** Comparison of cell and coccosphere sizes in cultured *C. braarudii* strains. Diameters of: **(a)** cells and **(b)** coccospheres ($\mu$m) as measured by light microscopy imaging. Both the HOL and HET phases have similar cell sizes. However, the HET phases have a significantly larger coccosphere size.

### 3.4 Photosynthesis and nutrient limitation

275     To test the response of nutrient limitation on photosynthetic response, we measured HET and HOL cultures under nitrogen replete and deplete conditions. During the nutrient repletion experiments, HET and HOL cells displayed similar photosynthetic





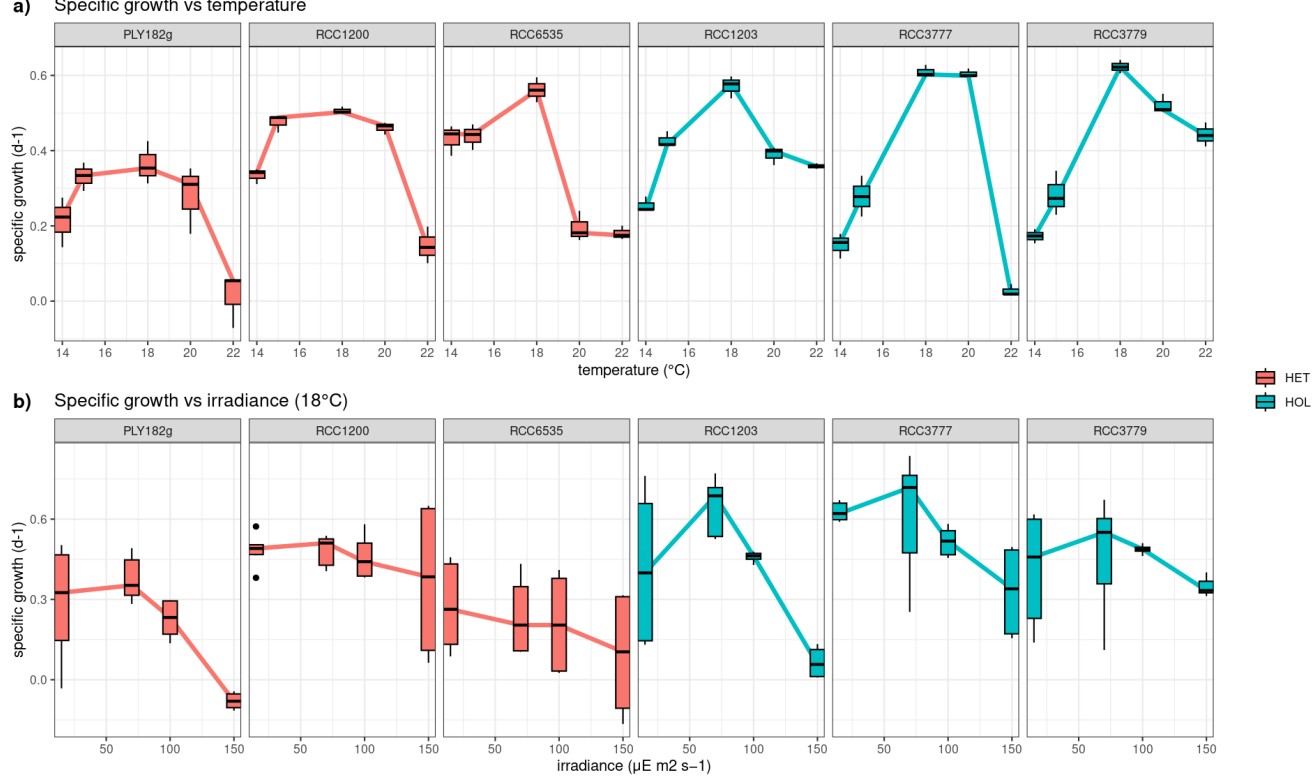

**Figure 4.** Temperature and Irradiance optima of *C. braarudii* HET and HOL strains. Optima for: (**a**) temperature and (**b**) irradiance. N=3 for each boxplot.

efficiency (Fv/Fm), (p-value = 0.056) and higher Electron Transport Rates (ETR; p-value = 0.004). When nutrient-limited, both HET and HOL strains exhibit a reduced Fv/Fm (Fig. 6). However, this inhibition is significantly higher during the nutrient-depleted conditions for the HOL strains than for the HET strains (p-value < 0.001).

280    Furthermore, the HOL phases show highly reduced ETR during photosynthesis, especially when exposed to high light. In contrast, HET strains displayed no or low reductions in ETR when exposed to high light (Fig. 6).

These results suggest that the HOL life cycle phase allocates less resources towards its photosynthetic machinery when nutrient-limited, reducing its photosynthetic efficiency and increasing its light inhibition sensitivity. This contrasts with the HET strains, which can maintain high photosynthetic efficiency when nutrient-depleted.

285 **3.5    Nitrogen and genome content**

We find that the HET life cycle phase contains twice the DNA content of HOL (Fig. A2), indicating that all strains have a similar genome size and ploidy drives the variations in genome content for the HET and HOL strains investigated in this study.




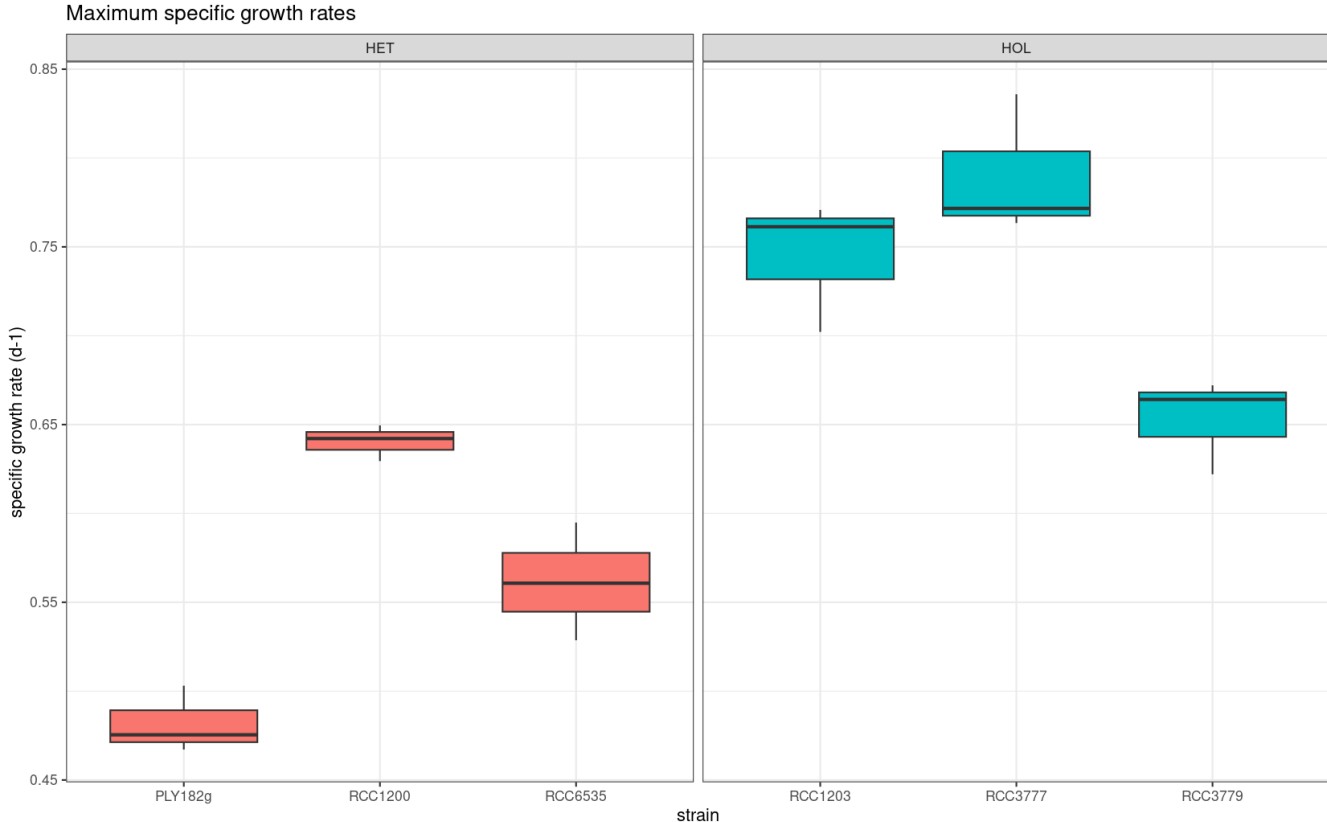

**Figure 5.** Maximum specific growth rates ($d^{-1}$) of *C. braarudii* strains at their respective temperature and light optima (N=3). Note that higher maximum specific growth rates are observed for HOL strains.

Under nutrient-repleted conditions (Fig. 7a), the genome content accounts for 5-15% ($10.7 \pm 5.8$ %) of the nitrogen budget, varying with the strains and with no significant difference between the phases (Fig. 7b). This percentage is similar to what is observed in diatoms, other algae, and cyanobacteria (Liefer et al., 2019; Geider and Roche, 2002).

However, under nutrient-depleted conditions, the genome content contribution towards the cell nitrogen budget stays the same in the haploid phase but increases in the diploid phase (20%; Fig. 7b). The genome content might thus be an important factor driving the different nutrient preferences observed between HOL and HET coccolithophore life cycle phases when nutrients are limited.

Notably, however, the minimum nitrogen quotas of the HET and HOL coccolithophore strains are very similar under nutrient-depleted conditions, with no significant difference between the observed HET and HOL cells (p-value = 0.985). Thus for HET to maintain a similar nitrogen quota under nutrient-depleted conditions, they must restrict other cellular resources. However, what part of the nitrogen demand HET strains reduce is not clear.





**Figure 6.** Response of photosynthetic machinery to nutrient limitation measured by pulse amplitude measurements (PAM). **(a-b)** Photosynthetic Efficiency (Fv/Fm); **(c-d)** Electron Transport Rate (ETR) in response to PAR.

## 3.6 Nutrient uptake rates and half-saturation constants

All strains show similar half-saturation constants (KN), maximum nutrient uptake rates ($V^{max}$), maximum growth rates ($\mu^{max}$), and minimum nutrient quotas ($Q^{min}$)(Table 1, Fig. A1). Furthermore, there is no apparent difference between the HET and HOL strains (Fig. A1).

The only noticeable difference observed in our nutrient parameter estimates are maximum nutrient quotas ($Q^{max}$), which was significantly higher for HET strains ($34.0 \pm 11.1$ pgN cell$^{-1}$) compared to HOL strains ($66.1 \pm 17.7$ pgN cell$^{-1}$, p-value $< 0.001$).



**a)** Nitrogen content



**b)** Relative DNA content

**Figure 7.** Nitrogen and genome content per strain under repleted and depleted nutrient conditions. **(a)** Cellular nitrogen quota, where depleted quota is equal to $Q^{max}$ and depleted quota is equal to $Q^{min}$; **(b)** DNA content relative to nitrogen content.

Our $Q^{max}$ measurements are similar to those reported in the literature. For PLY182g, we observe a PON of 54 pgN cell$^{-1}$, compared to a PON value to 40.6 pgN cell$^{-1}$ reported in Villiot et al. (2021). While for RCC1200, we observe a PON of 78.57 pgN cell$^{-1}$, compared to a value of 94.8 pgN cell$^{-1}$ in Gerecht et al. (2014).

While there is no direct comparison for KN, the KN value of 1.06 $\mu$M reported by Cermeño et al. (2011) for strain RCC1201 310 is similar to our estimates of 1.14 $\mu$M and 1.21 $\mu$M for PLY182g and RCC1200, respectively.





| strain | phase | $\mu^{max}$ | $Q^{min}$ | $Q^{max}$ | $V^{max}$ | $KN$ | $KQ$ |
|---|---|---|---|---|---|---|---|
| | | (d$^{-1}$) | (pgN cell$^{-1}$) | (pgN cell$^{-1}$) | ($\mu$mol cell$^{-1}$ d$^{-1}$) | ($\mu$mol l$^{-1}$) | (unitless) |
| RCC3777 | HOL | 0.83 | 18.45 | 21.43 | 16.75 | 1.50 | 6.20 |
| RCC3779 | HOL | 0.67 | 17.11 | 38.57 | 24.40 | 1.18 | 0.80 |
| RCC1203 | HOL | 0.77 | 27.56 | 42.14 | 24.39 | 0.96 | 1.89 |
| PLY182g | HET | 0.50 | 21.68 | 53.57 | 21.39 | 1.14 | 0.68 |
| RCC1200 | HET | 0.65 | 23.53 | 78.57 | 24.29 | 1.21 | 0.43 |

**Table 1.** Model parameters for the five strains tested.

### 3.7 Trait trade-offs

Overall, the only significant differences we observe between the HET and HOL strains are the coccosphere size, the maximum nutrient quota (Q$^{max}$) and the maximum growth rate ($\mu^{max}$). Following trait-based approaches, we argue that the trait trade-offs for *C. braarudii* likely relate to those three trait characteristics. Of these traits, both trade-offs between calcification status and growth and maximum nutrient quota and growth are plausible (Grover, 1991; Monteiro et al., 2016). However, as the influence of calcification status on growth is not noted in previous literature (Houdan et al., 2005), and we do not have PIC estimates of the HOL cells due to difficulties of making such estimates (Langer et al., 2022), we focus here on the implication of a maximum nutrient quota and growth rate trade-off.

### 3.8 Numerical simulation experiments

Our numerical simulations compared the competitive advantage of HET and HOL cells due to different relative maximum nutrient quota (Q$^{max}$) values as measured in our lab experiments (Fig. 8). We test the advantage of a higher Q$^{max}$ with no cost (Fig. 8a), and with reduced growth rate (0.5 and 0.7 d$^{-1}$ for the HET and HOL cells, respectively).

Both sensitivity model experiments show a clear advantage of having high Q$^{max}$ with intermittent nutrient supply with and without a growth rate cost (relative Q$^{max}$; Fig. 8). However, this advantage decreases when nutrient input is close to constant, or when the relative Q$^{max}$ is low and nutrient input is intermittent (Fig. 8b).

With a relative Q$^{max}$ between HET and HOL of *C. braarudii* of around 2 as measured in our lab study, the model shows that HET cells can be more competitive under nutrient intermittent regimes despite their lower observed growth rates.

### 4 Discussion

Our results show that higher nitrogen quotas and the ability to deal with varying pulsed nitrogen regimes may be a main bottom-up control of the distribution of *C. braarudii* life cycle phases, with diploid *C. braarudii* investing more into nitrogen storage at the cost of lower growth rates but with the benefit of being able to deal with more intermittent nitrogen supplies. This result suggests that lower nutrient intermittence during summer stratification rather than differences in nutrient concentration





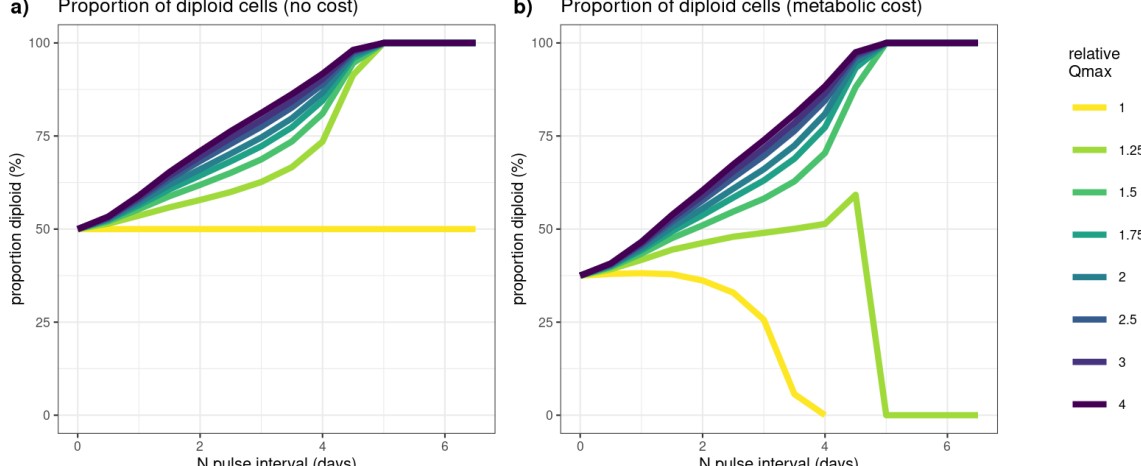

**Figure 8.** Sensitivity to $Q^{max}$ under varying nutrient pulses. **(a)** Assuming no cost (equal $\mu^{max}$); **(b)** Assuming a metabolic cost (HET = 0.5 $d^{-1}$ and HOL = 0.7 $d^{-1}$). relative $Q^{max}$ is the HET $Q^{max}$ relative to the HOL $Q^{max}$, where a value of 2 denotes a HET $Q^{max}$ twice the size of the HOL $Q^{max}$. The proportion of diploid cells is relative to the number of haploid cells.

could cause the in situ seasonal variation in HET and HOL coccolithophores observed in situ (de Vries et al., 2021), including *C. braarudii* (Malinverno et al., 2009; D'Amario et al., 2017).

Previous studies also identified nutrient storage as an important driver of phytoplankton distribution in the modern ocean (Falkowski and Oliver, 2007; Grover, 1991), with turbulence affecting nutrient inputs (Falkowski and Oliver, 2007; Grover, 1991; Tozzi et al., 2004).

     Besides, nutrient storage relates to functional types, with a succession of nutrient storage ability observed for diatoms, coccolithophores and dinoflagellates (Margalef, 1978). For example, diatoms boast highly specialised nutrient storage vacuoles

compared to dinoflagellates (Lomas and Glibert, 2000), which allows diatoms to exploit intermittent nutrient regimes better. Furthermore, differences in nutrient storage may influence the evolution and distribution of diatoms and coccolithophores on geological time scales (Tozzi et al., 2004). For instance, the early Panthalassa (250-190 Ma) and associated lower global turbulence seem to have favoured coccolithophores, while in the modern day, greater turbulence globally favours diatoms (Tozzi et al., 2004).

The potential role nutrient storage and turbulence play in determining phytoplankton distribution across geological time scales are interesting when contextualised with the idea that the haplo-diplontic life cycle of coccolithophores allows single species to expand their niche in the modern ocean (de Vries et al., 2021). This concept suggests that the coccolithophore life cycle might also prevent extinction across geological time scales, allowing adaptation toward a larger range of nutrient regimes. Differences in turbulence preference are also notable in the context of coccolithophore success during anthropogenic climate

change, as climate models predict increased stratification in our near future oceans (Fu et al., 2016)



The advantage of higher nutrient quotas comes from the ability to withstand more extended periods without nutrients and extended maximum uptake rates, as it takes longer for the quota to be satiated. Previous works by Verdy et al. (2009) and Grover (1991) suggest that a positive relationship between maximum uptake rates and higher nutrient quotas primarily drives this. However, here we illustrate that this effect occurs even when is similar, as even with the same values, it takes longer for
the quota to reach maximum values if is high.

Besides HET's higher nitrogen storage ability, we furthermore observe significantly reduced photosynthetic ability for HOL cells in nitrogen-depleted conditions compared to HET cells. Although we do not explicitly model this effect, this could further differentiate the competitiveness of HOL coccolithophores when experiencing intermittent nitrogen supplies.

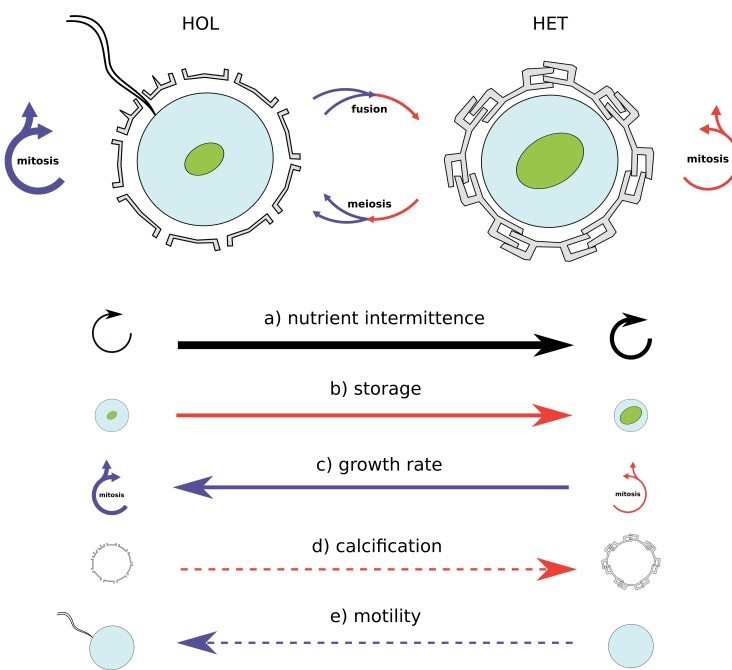

**Figure 9.** Nitrogen intermittence trait trade-offs of the *C. braarudii* life cycle. This study identifies a nitrogen quota and growth trade-off as a key driver for differences in HET and HOL *C. braarudii* distribution. While the investment of HET cells into comparatively higher nitrogen storage comes at the cost of lower growth rates, this disadvantage is overcome when nitrogen supply is intermittent. Likewise, HOL cells out-compete HET cells when the nitrogen supply is constant due to higher growth rates but are out-competed when the nitrogen supply becomes intermittent. Other potential competitive advantages under different nutrient regimes include calcification state (d) and motility (e), which also vary between HET and HOL cells.

While our work focuses on nitrogen storage and the subsequent ability to deal with different nitrogen-input regimes, other trait differences in HET and HOL are closely related to this trade-off. For example, other traits that increase HOL competitive-
ness in highly stratified regions include motility (Frada et al., 2018) and lower sinking rates (Bach et al., 2012). Furthermore,





higher grazing protection of more heavily calcified HET life cycle phases might also be closely related to turbulence, as highly turbulent regimes are associated with higher predator encounters (Kiørboe, 1997). Phagotrophy might furthermore offset nutrient requirements for the HOL life cycle phase in nutrient-limited conditions (Rokitta et al., 2011; Avrahami and Frada, 2020),

although no notable difference between haploid and diploid *E. huxleyi* or *C. leptoporus* are observed (Avrahami and Frada, 2020).

Because these other traits are difficult to constrain based on laboratory data, further model-based sensitivity analysis of such trait trade-offs in more complex ecosystem models could be useful in terms of determining the role some of these other traits play. For example, in the context of our lab results, the impact of coccosphere size on grazing susceptibility without accounting

for additional predator-prey effects could already be very informative in determining the role this might play on distribution dynamics.

Our study shows that the genome is a significant part of both the HET and HOL nitrogen budget when cells are nutrient deplete, with the relative contribution of the HOL genome twice that of the HET genome. Nonetheless, the nitrogen quotas of both phases are similar when nitrogen deplete, which suggests that other trade-offs not considered here might be essential

in sustaining similar minimum nitrogen quotas for HET and HOL cells. However, a more thorough macromolecular analysis is necessary to elucidate which budgets diploid cells are reducing. For this, proteomics would be particularly useful as it constitutes most of the nitrogen budget of phytoplankton (Liefer et al., 2019).

Furthermore, our research focuses primarily on nitrogen storage and uptake dynamics, but phosphate is also critical in regulating global phytoplankton distribution (Tyrrell, 1999). The role of phosphate storage plays in HET-HOL competitive

dynamics thus also warrants further research, particularly as, unlike nitrogen, phytoplankton cells utilise phosphate-specific storage bodies in the form of polyphosphate (Liefer et al., 2019), which for coccolithophores is tied to calcium storage (Sviben et al., 2016; Gal et al., 2018).

## 5   Conclusions

Using an integrated approach featuring both laboratory experiments and numerical simulations, we find that *C. braarudii* life

cycle phases have different nitrogen storage and maximum growth rates, but similar light preference, nitrogen uptake rates and temperature optima. The maximum growth is inversely related to nitrogen storage, with HOL having higher growth rates and lower nitrogen storage and HET having lower growth rates and higher nutrient storage. This result indicates a potential trade-off between growth and nutrient storage, which could be a primary bottom-up control of *C. braarudii* life cycle phase distribution.

Higher nitrogen storage allows HET coccolithophores to better exploit regimes of intermittent nutrient supply, such as highly turbulent regions. This benefit comes at the cost of a lower competitive ability due to a lower maximum growth rate when turbulence is low, and nitrogen supply is constant.

The *C. braarudii* HET life cycle phase's ability to better store nitrogen is also associated with high photosynthetic efficiency when the cell experiences nitrogen depletion further suggesting an important role of photosynthesis during nitrogen depletion.




We found that genome content plays a minor role in the nitrogen budget of nutrient-repleted cells but is relevant for nitrogen-depleted HET cells, where DNA attributes ≈20% of the nitrogen budget. Notably, this higher genome content does not correlate with higher minimum nitrogen quotas for HET cells, suggesting that HET cells reduce other cellular components to maintain similar minimum nitrogen quotas. However, as we can not infer such budget cuts, our work highlights that more thorough macromolecular investigations of the HET-HOL life cycle phases are warranted.

Overall, our work shows the advantage of studying the coccolithophore life cycle in a trait-based framework. By comparing differences in traits between the different life cycle phases, we identify the key traits that determine the distribution of coccolithophore phases. This work furthermore provides parameterisations needed for more in-depth numerical simulations and analysis, which provides an exciting avenue for future coccolithophore life cycle research.

*Code availability.* The HET-HOL competition model is available on figshare: 10.6084/m9.figshare.22717873.

*Author contributions.* JdV, GW and FM conceptualized the manuscript. JdV, GW and GL conceptualized the lab experiments. JdV and FM conceptualized the numerical simulations. JdV performed the lab experiments, developed the competition model, conducted the formal analysis, and visualized the results. JdV, GW, FM, GL and CB interpreted the results. JdV, GW and FM prepared the manuscript with contributions from all co-authors.

*Competing interests.* The authors declare that they have no conflict of interest.

*Acknowledgements.* This research has been supported by a NERC GW4+ DTP studentship (grant nos. NE/L002434/1) to JDV, the National Environmental Research Council grant nos. NE/X001261/1 to FM and NE/N011708/1 to GW; the European Research Council (SEACELLS, grant no. 670390) to CB; and funding from the Spanish Ministry of Education through a Maria Zambrano grant to GL.



# Appendix A

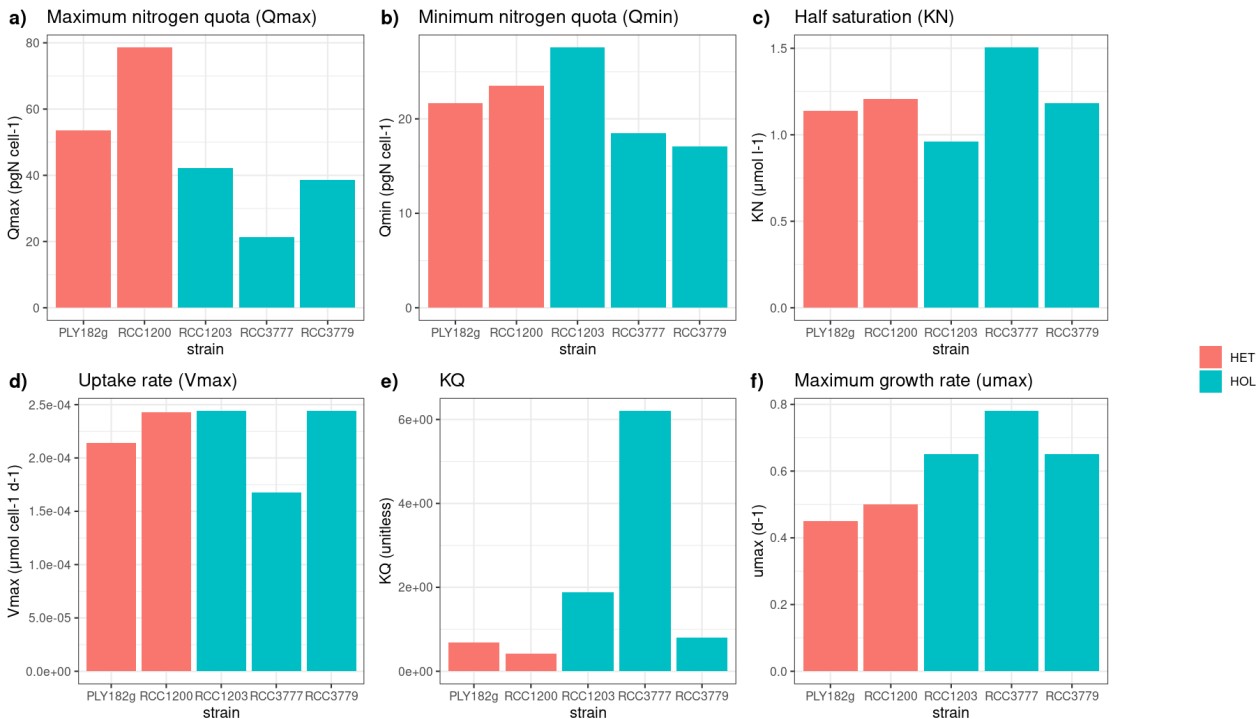

**Figure A1.** Estimated and measured model parameters. **(a)** maximum nitrogen quota; **(b)** minimum nitrogen quota; **(c)** half saturation constant; **(d)** uptake rate; **(e)** $KQ$; **(f)** maximum growth rate.



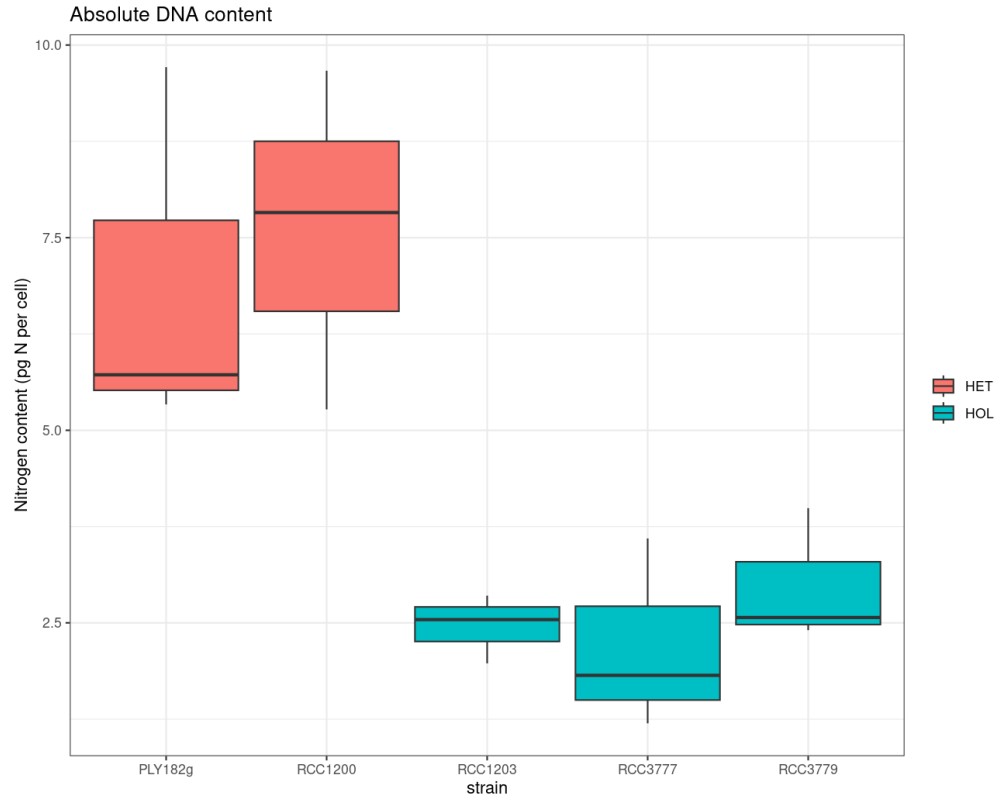

**Figure A2.** Nitrogen content of the different HET and HOL *C. braarudii* strains.





| Experiment | $\mu^{max}$ HET | $\mu^{max}$ HOL | $Q^{max}$ HET | $Q^{max}$ HOL | relative $Q^{max}$ | pulse intervals |
|---|---|---|---|---|---|---|
| | (d$^{-1}$) | (d$^{-1}$) | (pgN cell$^{-1}$) | (pgN cell$^{-1}$) | (unitless) | (d) |
| No cost | 0.7 | 0.7 | 30 | 30 | 1 | (0.5, 1, ..., 6) |
| No cost | 0.7 | 0.7 | 37.5 | 30 | 1.25 | (0.5, 1, ..., 6) |
| No cost | 0.7 | 0.7 | 45 | 30 | 1.5 | (0.5, 1, ..., 6) |
| No cost | 0.7 | 0.7 | 52.5 | 30 | 1.75 | (0.5, 1, ..., 6) |
| No cost | 0.7 | 0.7 | 60 | 30 | 2 | (0.5, 1, ..., 6) |
| No cost | 0.7 | 0.7 | 90 | 30 | 3 | (0.5, 1, ..., 6) |
| No cost | 0.7 | 0.7 | 120 | 30 | 4 | (0.5, 1, ..., 6) |
| Metabolic cost | 0.5 | 0.7 | 30 | 30 | 1 | (0.5, 1, ..., 6) |
| Metabolic cost | 0.5 | 0.7 | 37.5 | 30 | 1.25 | (0.5, 1, ..., 6) |
| Metabolic cost | 0.5 | 0.7 | 45 | 30 | 1.5 | (0.5, 1, ..., 6) |
| Metabolic cost | 0.5 | 0.7 | 52.5 | 30 | 1.75 | (0.5, 1, ..., 6) |
| Metabolic cost | 0.5 | 0.7 | 60 | 30 | 2 | (0.5, 1, ..., 6) |
| Metabolic cost | 0.5 | 0.7 | 90 | 30 | 3 | (0.5, 1, ..., 6) |
| Metabolic cost | 0.5 | 0.7 | 120 | 30 | 4 | (0.5, 1, ..., 6) |

**Table A1.** Competition model experiments. Competition between HET and HOL cells was simulated with and without a metabolic cost (reduced $\mu^{max}$). For each nutrient pulse interval, different ratios of HET and HOL $Q^{max}$ (relative $Q^{max}$) were tested.



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
