# Peer review of "A critical trade-off between nitrogen quota and growth allows Coccolithus braarudii life cycle phases' to exploit varying environment"

_EGUsphere, 2023_

## Author Comment (AC1)

**BG Discussion: Reply to RC1**

We would like to thank the first anonymous reviewer for the kind and constructive feedback. A detailed response to their comments is found below.

**Overview:**

de Vries et al. present research on the distinct life cycle of *Coccolithus braarudii*, a species of coccolithophores, focusing on the trade-offs between its haploid and diploid phases. They examined how these phases respond to environmental factors such as light, temperature and nutrients. With the laboratory experiments they showed similarities in cell size, nitrogen requirements, uptake rates, and optimal temperature and light conditions between the phases. However, differences were noted in coccosphere size, maximum growth rates, and nitrogen quotas. Authors offered the explanation that trade-off is observed between maximum growth rate and nitrogen quota, with the haploid phase favouring higher growth rates and lower nitrogen storage, while the diploid phase shows the opposite pattern. They also run model simulations that indicated that trade-off allows *C. braarudii* to more effectively utilise varying nitrogen conditions. The ability of the diploid phase to store more nitrogen proves beneficial when nitrogen is intermittently available, while the higher growth rate of the haploid phase is advantageous when nitrogen is constantly available. This study suggests that the trade-off between nitrogen storage and maximum growth rate is a critical factor determining the distribution and functionality of the C. braarudii life cycle.

The results of this study contribute to understanding of how haplo-diplontic life cycle works and I believe it is an important addition to the scientific literature. The paper introduces the concept of trade-off between maximum growth rate and nitrogen quota in the haploid and diploid phases of *C. braarudii*, which has not been identified previously and the models used to test the tradeoffs under different environmental conditions also provide new tools for the field.

However, I do have some major and minor comments and concerns that are provided below. In terms of the writing, it is recommended to revise both the results section and subsequent discussion. The results section currently contains significant conclusions and comparisons, going beyond the objective presentation of the findings. Therefore, it would be beneficial to adjust the content to focus on a more objective reporting of the results. Similarly, the discussion section should be modified to reflect a balanced and unbiased analysis rather than incorporating conclusions that extend beyond the presented data.

**Main remarks:**

Abstract

Considering the strong photoinhibition observed in figure 6b under depleted conditions, the authors' claim that HOLs prefer high light and low nutrients becomes questionable. The pronounced differences between high/low nutrient conditions and between HOL/HET phases are noteworthy enough to merit inclusion in the abstract.

The high light/low nutrient line "These life cycle phases vary significantly in inorganic carbon content and morphology, and inhabit distinct niches, with haploids generally preferring low-nutrient and high-temperature and light environments." refers to their environmental niche, we will update the phrasing to reflect this more clearly.

**Introduction**

Ls 276, 277, 278: The use of the broad term "nutrient" instead of the specific term "nitrogen" throughout the paper may lead to potential confusion. This generalisation might hold true for nitrogen but not necessarily for other nutrients such as phosphorus. I recommend considering the use of "nitrogen-depleted" specifically in the results section, where other nutrients remain present, and then extrapolating to the term "nutrients" in the discussion to encompass a broader context.

**We will replace nutrient with dissolved inorganic nitrogen as suggested by the reviewer**

L 135-136: The assumption made regarding DNA content is a bit concerning. It is reasonable only if the G2 phase is significantly longer than the G1 phase during the cell cycle and if the stationary phase primarily consists of G1 cells. It is essential to have supporting information to justify this assumption. Without such information, any calculation could potentially be true. For example, if the stationary phase primarily consists of cells in G2, the average DNA content could be even higher than that of a dividing population. Furthermore, if the G1 phase is longer than G2, the situation becomes even more complex. It is important to thoroughly discuss this issue and, if uncertainty exists, take it into account in both the assumption and subsequent models.

We agree that the assumption that DNA content is different in stationary phase cells is not supported by experimental data. We will therefore remove the assumption that DNA content changes from our calculations, recalculate the values and update the conclusion. We will also include some discussion to highlight that we haven't included assumptions about shifts between about G1 and G2 phase that could influence the result. However, we should also point out that this wouldn't change our conclusions despite the magnitude of the difference.

L 280: Please exclude the latter part of the sentence: "Furthermore, the HOL phases show highly reduced ETR during photosynthesis, especially when exposed to high light." This statement holds true only under depleted conditions.

**We will update this as suggested.**

**Figure 1**

In Figure 1e, the chromosomes of the HET phase represent metaphase chromosomes, consisting of two chromatids, indicating a genome still at the haploid state (n). The 2n condition, on the other hand, would typically represent pairs of chromosomes, which can be illustrated side by side but should not be linked by their centromeres.

**The representation of chromosomes in the figure is a minor detail, but something we can fix.**

Considering the abundance of figures in the paper, it might be appropriate to consider moving the entire Figure 1 to supplemental material, as Figure 9 provides more comprehensive information. Additionally, certain differences observed in Figure 1, such as cell size, contradict the observations, which could potentially lead to confusion or misinterpretation.

**We will move Fig 1 to the supplement.**

**Material and methods**

L 9: Regarding cell size, was the difference between the presence and absence of a coccolith within the cell considered? Were the cell sizes estimated throughout experiments? During the exponential phase, significant coccolith calcification occurs, which could potentially impact the conclusions made. It would be valuable to investigate and account for any potential influence of coccolith presence on cell size when drawing conclusions. Additionally, please provide information on how many cells were counted per image/strain/condition?

We didn't consider whether internal coccoliths contribute to cell volume. However, it is clear from previous experimental manipulations that even when calcification is inhibited cells usually contain an internal coccolith (coccolith formation is paused, but the immature coccolith is retained). So whilst we haven't compared internal coccoliths in exponential and stationary phase cells directly, we wouldn't expect calcification rate to have a direct impact on cell size. For the purpose of the model it is important that we include a measurement of cell size. Whilst calcification could influence this parameter, it does not impact our conclusions.

At least 100 cells were counted per strain. We will update the methods to reflect this.

L 118: Although I am not an expert in quotas, I find it surprising that the estimation of Qmax and Qmin is based on completely distinct methods. Please provide the specific equations used for calculating Qmax and Qmin in the given context, especially as these are a major part of modelling efforts.

We agree that using CN for Qmin would have been ideal, but a previous comparison by Perin et al., 2008 demonstrated that these two methods give very similar results.

We will specify the equations for clarity.

L 125-129: The values of Fv/Fm and ETR are "computed" rather than directly measured, particularly ETR, which relies on parameters that are subject to certain hypotheses, such as the efficiency of light capture (cell concentration, antenna size, etc.). While it is appropriate to present the differences in ETR in the results section, the underlying cause and significance of these differences should be thoroughly discussed.

**We will add this discussion.**

L 126: In section 2.2 on nutrient limitation, it is mentioned that the "deplete" concentration is 20  $\mu$ M. Therefore, the condition indicated as 220.5  $\mu$ M should be referred to as "replete." The same clarification should be made in the legend of Figure 7.

**This was a typo, thanks for spotting it. We will update this.**

**Results**.**

Several portions of the text extensively discuss the findings and would be better placed in the discussion section. While not explicitly mentioning all of them, it is crucial for the authors to consider that the results section should solely describe the outcomes of the data analysis. Conclusions, comparisons, and in-depth discussions should be reserved for the dedicated discussion section.

L 251-254: This paragraph belongs in the discussion section.

We will do this

L 253: Cell surface (no "'s ")

We will fix this

L257: Duplicate: cycle cycle

**We will fix this**

L 259: The phrase "unlike temperature" is awkwardly worded. Both for temperature and light, the differences are not life cycle phase-specific.

**We will remove "unlike temperature"**

L 263-269 This section belongs in the discussion section.

L 270-271 This sentence belongs in the discussion section.

**We will move 263-271 to the discussion**

**Figure 3**: Could you please provide the information on the number of cells counted for each condition in this section here? Additionally, remove the conclusions "Both the HOL and HET ..." from the figure legend and especially if mentioning significantly provide the appropriate data supporting this wording. It would be more appropriate to include a statistical test and describe that.

We will add the number of cells counted and the p-value for the significance statement. We feel the text helps with the interpretation of the figure, so we have retained the text "Both the HOL and HET ..." but clarified the nature of the statistical tests that support these conclusions.

L 276: Please insert "maximal", as in: "similar maximal photosynthetic efficiency"

**We can update this**

**L 286: What is "Fig. A2" ?**

The caption of Fig. A2 should read "absolute DNA content (pg N per cell)" we will update this.

L 292-294: This sentence belongs in the discussion section. In addition: considering the DNA content and its potential "cost," I find it very speculative, as evolutionary pressure in such cases would likely lead to the development of more compact genomes. However, it appears that this is not the case here, and therefore, the connection between DNA content and its cost might be somewhat far-fetched. - Discuss.

We will move this to the discussion and contextualize it with previous studies. However, we disagree that this is too far-fetched. Nitrogen requirement is a strong selector for phytoplankton fitness in oligotrophic regions, and there are trade-offs associated with having a very compact genome. Given that the genome contributes substantially to the total cellular nitrogen budget, it's reasonable to assume that switching between life cycle phases may help to lower the minimal N quota.

We agree that genomic reduction would be an effective alternative strategy to lower N quotas, although this is primarily seen in organisms adapted to ultra-oligotrophic regions. In an organism exhibiting a haplo-diplontic life cycle that inhabits a much wider range of nutrient regimes, it's important to consider how genome reduction would influence both life cycle phases.

Interestingly, there is evidence of genome reduction (or at least extensive gene loss) in the HET phase of some coccolithophores. Multiple environmental isolates of Emiliania huxleyi

have lost multiple genes associated with the HOL life cycle phase and are therefore stuck in the HET phase (von Dassow 2015 ISME). Moreover, this gene loss was observed primarily in oligotrophic isolates. So genomic reduction could be an alternative strategy for lowering N quotas in coccolithophores in oligotrophic regions, but with the major consequence of the loss of sexual reproduction.

**Figure 5**

The summarizing Figure 5 is interesting. It would be helpful to include the conditions associated with each maxima.

**This is a good suggestion and something we could add.**

Figure 6:

a) Consider renaming Fv/Fm as "Maximal yield" or a similar term, as it represents an estimation of the maximal yield, but theoretical, since it is measured in the "dark"(F0).

**We are happy to rename Fv/Fm with maximal yield as suggested.**

b) Remove "light inhibition" from the title.

**We will do this.**

Clarify whether the average values presented are based on all HET/HOL strains or just one of each. This information is important.

**It is the average. We will update this.**

Provide details on the duration of cell exposure to each light level. Are different samples used at each level?

**We will add this information to the methods section.**

In Figure 6b, ensure consistency with the positioning of deplete and replete conditions. Currently, they are presented on the left and right sides in (a) but are opposite in (b), which could be misleading.

This is a good point and we will update the figure.

**L 301-302, what is "Figure A1"?**

We are unsure about this comment, but figure A1 is part of the appendix

Additionally, the results of Table 1 and Figure 8 are briefly described and may benefit from a more comprehensive explanation.

We can expand the captions, especially for Table 1.

**Discussion**

Please revise the write-up, with the intention of systematically integrating the parts that were previously discussed in the results section.

We will integrate the sections the reviewer suggested should be moved into the discussion.

---

## Author Comment (AC2)

**BG Discussion: Reply to RC2**

We would like to thank the second anonymous reviewer for the in-depth and constructive feedback.  A detailed response to questions raised are provided below.

**General Comments:**

This manuscript considers the trait trade-offs of the haploid and diploid life phases of Coccolithophores under different environmental conditions (e.g.changes in light, nitrogen and temperature) and provides important contributions to the literature.

HOL type strains appear more sensitive to nitrogen stress (in terms of Fv/Fm and ETR), although under replete conditions HOL strains have higher max growth rates/Fv/Fm/ETR. While HOL strains have higher max growth rates, their nitrogen content per cell is lower under N replete conditions. Nitrogen deplete conditions reduce N content in both life cycle types to similar levels, but the allocation of N quota to DNA content is increased in HET strains.

Experiments using a chemostat model are used to determine if trait-tradeoffs confer competitive advantages to HOL vs HET under varying nutrient supply. The authors conclude that higher growth rates and smaller nitrogen storage seen in the haploid phase are advantageous under consistent nitrogen supply, while larger Qmax with lower growth rates is advantageous under conditions of sporadic nitrogen input. This work has applicability to informing ecosystem models with life cycle distinctions in coccolithophore populations.

Overall, this manuscript presents data comparing coccolithophore stains and life cycle types under difference environmental conditions, and shows that there are clear differences in the physiological response and growth advantages of HOL vs HET lifestyles. Furthermore, modeling these trait differences provides a unique tool for broader applicability across time and space.

However, the photosynthetic response data needs more explanation and discussion, as low Fv/Fm can imply nutrient stress but is sensitive to nutrient history and length of acclimation to new nutrient regimes. Further detail of culture methods during these experiments would be helpful. The chemostat modeling experiments also require

clarification regarding the sensitivity experimental methods (not the model construction), and better explanation of the results when presenting Fig. 8.

We will improve the explanation and discussion of the photosynthetic response and chemostat model as described in the response to the specific comments below.

Figures 1 and 9 are beautiful, but may contain more than the scope of the data presented in this paper. And in a similar sense, the discussion section should remain more closely linked to data presented in the results section. Some comments on other related topics are appropriate at the end of the section, but should be minor additions to the core discussion of the data presented.

We will move Fig 1 to the supplement and clarify the caption of Fig 9 to clarify which traits are discussion points and which traits are supported by our results.

**Specific Comments:**

Line 10/11: What is the difference between nitrogen requirement and nitrogen quota?

Nitrogen requirement refers here to qmin and nitrogen quota to qmax. However, as technically both Qmin and Qmax are nitrogen quotas we will update this.

Line 14: what model? Hasn't been introduced yet...

We will clarify this.

Line 41/43/47: C. braarudii, already stated full name in line 37

We will update.

Line 53: Match body text and figure text more closely? ie genome N content,  transport proteins (nitrogen uptake?)...

We will standardize phrasing better.

Line 60: ...for changes in coccolithophore traits in response to light, nutrients....

We will update.

Line 67: Section 2.1 and 2.2 are both experimental?: perhaps 2.1 is specified as strain comparison experiments and 2.2 is nutrient limitation experiments? Or combine all the culture conditions under one section here? This feels like a helpful clarification for when you present the results that switch back and forth between comparing differences in strains, life-cycle phases and nutrient status...

We will rephrase the two sections:

1. Strains and culture conditions
2. Nutrient limitation experiments

Line 74-77: Does the use of HOL vs HET strains isolated from mostly coastal locations influence the ability to fully understand "low nutrient/high light/stratification adapted" vs "turbulence adapted" life cycles? i.e. how do these strains compare to more open ocean strains? and is there comparison of light cycles collected from a single location?

Yes, this is a good point. This is a clear limitation as we are limited by the strains available to us. We will add this consideration to the discussion

Line 78: It may be helpful to use "nitrogen" in place to "nutrient" to clearly specify that these experiments are nitrogen limitation experiments. Perhaps the model equations below could remain generalized to any "nutrient", but once your data is applied, I think using "nitrogen" may be more appropriate/specific.

This also came up in review 1 and we will replace nutrient with nitrogen.

Line 80: 20uM NO3 is usually a quite adequate amount of N in the surface ocean, where max concentrations may only reach 35uM below the nitracline, and could be considered replete in other studies. I'm guessing early exponential nitrogen growth physiology would be similar in both your deplete and replete conditions and differences would be seen in a batch culture only once nitrogen begins to be much lower.

The reviewer is correct that during exponential growth phase both experiments are identical. We do not take measurements for the deplete culture conditions until growth rate has ceases (i.e., at which point nitrogen is depleted to close to 0 uM NO3). We will clarify this in the text.

Line 82: Triplicate bottles? Culture volumes? Vessel type? temperature control method? More detail on culture methods would be helpful.

We apologise for not including this fundamental information and will include it in the new text. All experiments were conducted in triplicate with 50ml tissue culture flasks with 20ml of volume. All experiments were done inside a temperature-controlled room.

Line 95: BMG?

BMG is the manufacturer we will rephrase:

"fluorescence microplate (BMG Labtech)"

Line 102/192: It would be nice to see that raw cell density data over time in the supplement/appendix... how many days does it take to get to 10-20k cells?

Good suggestion, we can include this in the supplement.

Line 120: $Q^{max}$ is also divided by cell count? $Q^{min}$ calculation uses initial media nitrogen concentration? Unclear

Qmax is also per cell and Qmin uses initial media nitrogen concentration. We will clarify this.

Line 125: Citation for fluorescence. Parkhill 2001? Falkowski?

We will add Maxwell and Johnson JXBot 2000 as the citation for fluorescence

Line 136: Is there further explanation of this assumption or a citation? Assuming half the DNA content is assuming ALL replete cells are actively in the DNA replication stage…?.

We will remove this assumption as discussed in the reply to the other referee.

Line 163: units for P and N and Q

We will add this

Line 167:  is $Q^N$ the same as Q? Specify if N in equations is for nutrients or nitrogen.

Yes, we will fix this

Line 169: Flynn 2002?

The reviewer is correct that this was initially proposed in Flynn et al., 2002, we will update this.

Line 200: What is $m^{inf}$ ?

Mu infinity is the maximum theoretical growth rate (as opposed to the realized maximum growth rate). We will explain this in the new manuscript.

Line 213: Is *E.hux* spelled out earlier in manuscript?

No, we will update it to the full name.

Line 226: mortality term (m, + units)

We will add this.

Line 228: redefine terms that have i subscripts (?), or define i once at the start of the chemostat model section.

We will define i as suggested (it refers to the different strains in the chemostat model)

Line 237: "Using the chemostat model, we…."

We will update this.

Line 237: does this mean main difference between experimental nutrient treatments?

This refers to the main difference for all experiments (I.e. nutrients, and light and temperature optima) but not PAM measurements. We will clarify this.

Line 238-242: Some of the justification and theory of this model experiment may be better in the discussion section? Then, this model experiment needs more explaination. I am having trouble interpreting Fig. 8 when I don't know how long the model was run, what an "input scenario" is or how you determine and implement "metabolic cost" differences.

Each model was run until steady state. Nitrogen input was constant and above saturation, although the exact number did not impact relative contributions.

Metabolic cost was defined as a lower growth rate for HET (0.5) relative to HOL (0.7). This number was based on the observed difference in maximum growth rate. We will clarify this in the Methods section of the final manuscript.

We will keep the description in the method so readers can better understand Figure 8.

Line 248: The cell size and coccosphere size were measured in replete and depleted nitrogen conditions (methods), so is this showing average across all nitrogen treatments? Or is this under culture maintenance conditions (15C + 50 mE m-$^{-2}$s$^{-1}$) or respective temp and light optima? Clarify

This is under maintenance conditions. We will clarify this.

Line 253: Would a larger coccosphere volume have a higher metabolic cost for production? It may not influence uptake rates, but would it possibly influence growth rates in addition to grazing susceptibility?

Yes, this is a possibility as discussed in line 315

Line 259: Are average maximum growth rates observed between HET and HOL strains statistically different? Sort of looks like HOL growth rates have higher maxima within your treatment ranges for both temp and light.... (oops, addressed in next section!)

Line 262: Was there a reason for not testing higher irradiances in the lab? Eg. artificial lighting can only reach 150?

We did try higher irradiances but had trouble removing temperature effects

Fig.4: Not sure box plots are appropriate for n=3??

This is a valid point. We will update the plot to show means with a standard deviation error bar.

Line 275: Are the Fv/Fm tests conducted in the first transfer generation into N depleted media? I think there are difference in observable change in Fv/Fm based on transient

changes in N supply vs balanced growth when cells are fully acclimated to low N supply....
(Parkhill?)

For the "deplete" culture conditions the strains were maintained at exponential phase under nutrient replete conditions and then inoculated in media which contained nitrogen concentrations which were initially non-limiting, but low enough to allow the full draw down of nitrogen once the cultures reached stationary phase.

Fv/Fm was then measured at stationary phase to represent "deplete" culture conditions and measured at exponential phase to represent "replete" culture conditions.

The experiments were conducted in batch culture, so we were unable able to effects of transient changes in nutrient concentration.

Line 282: *fewer* resources? Is Fv/Fm measured during exponential growth phase? I don't think changes in Fv/Fm due to nutrient stress occur until very low level of N....? Are lower growth rates in HOL relevant to the higher Fv/Fm and ETR measured in N depleted cultures?

Fv/Fm for the replete cultures were measured during stationary phase with very low levels of nitrogen.

Fig. 6 put Depleted and Replete panels on consistent side in both a and b

We will fix this.

Line 300: What do you mean by "no apparent difference"? KQ and mmax bars look different in A1....show stats of some kind? I also still want to see the raw cell abundance data that these growth rates were estimated from...

We will rephrase to: Furthermore, there is no apparent difference between the HET and HOL life cycle phases (Fig. A1).

Fig 7. *replete* quota is $Q^{max}$

Thank you for the correction. We will fix this.

Lines 306-310: I think these comparisons to the literature should be in the discussion section.

We are happy to move this.

Table 1: Order strains in consistent order with other figures. Is this caption correct? I thought Table A1 had the model parameters used. This table is the parameter value estimates from your lab data?

We will update the captions to be consistent with both the table and figure. The model parameters and the measured/estimated parameters have the same values. However, some are represented with different units, we will fix this for clarity.

Line 311: "Trait trade-offs" section feels like discussion.

We include this discussion in the methods section as it is important to understand the methods used for the model.

Line 320: I think you are saying "competitive advantage" to mean the relative abundances of HOL vs HET cells? But it is unclear. This section needs better explanation of the results (possibly clarification in the methods section too). How much nitrogen is in a pulse? Does it go to zero between pulses? What is relative Qmax? How much nitrogen is supplied under continuous supply? Low nitrogen concentrations should also influence abundances of low vs high growth rate strains, I don't see how this is ruled out by your study.

The competitive advantage is the ratio of the HET and HOL cells. We will clarify this.

To answer the other comments: we used a nitrogen pulse with concentration above saturation. Running the experiment with a different concentration will not change our final results and conclusions. This is because the nitrogen concentration affects the absolute abundance of each strain, but not their relative abundances since the values of half saturation constant and maximum uptake rates are the same for both strains.

Between pulses the nitrogen input is zero, which leads the nitrogen concentration to reach equilibrium at zero as well.

The relative Qmax is already defined in the methods. It is the ratio of the Haploid to Diploid Qmax. We clarified this further in the results.

Line 345: Turbulence is not really part of this dataset...

Turbulence and nutrient intermittence are used interchangeably here following several of the cited authors. However, we can add some discussion around this.

Line 352: What's an "extended maximum uptake rate"?

We can rephrase this to: "Longer periods at which maximum uptake rate is sustained. "

Line 354: When *what* is similar?

Nutrient uptake rates. This will be fixed

Line 356: Explain what Fv/Fm and ETR tell you about photosynthetic ability either here or in the Fv./Fm/ETR methods section.

We can add further details in the methods section

Figure 9. Are motility and calcification data part of this paper?

No but they co-occur and are thus illustrated as dotted lines. We will further clarify this.

"-a" twice in second caption sentence.

 Thanks we will fix this

**Technical Corrections:**

We will fix all the corrections as suggested below by the reviewer.

Line 79: "…modifying the K/2 media from an initial $NO_3$ concentration of 220.5 mM down to 20 mM."

Line 86: no new paragraph

Line 89: "Cell size was measured for each stain in both nutrient replete and nutrient depleted cultures"

Line 95: "…growth rates were estimated using change in cell abundance over time as estimated using…"

or change "growth rate" to "cell abundance" and leave end of paragraph as is.

Line 112: temperate-sensitivity

Line 125: Walz WATER-PAM (?) Pulse-amplitude modulated

Line 125: nutrient replete and nutrient depleted

Line 126: PAM is not a measurement "…nitrogen replete experimental cultures grown with 220 mM $NO_3$, Fv/Fm and ETR were measured…."

Line 127: "…For the nitrogen depleted experimental cultures,"…….."and Fv/Fm and ETR were measured once cells…"

Line 135: replete

Line 181/185/189/205/210/etc.: subscript $K_N$ in equation and text

Line 241: -it

Line 225: phytoplankton abundance of both HET and HOL strains

Line 275: which temp and light conditions are the nutrient experiments conducted at, assuming respective temp and light optima, but unclear? Fig. 6b is showing the average of both HET and HOL strains...

Line 276: depleted, replete

Fig 6 – why are RCC1200 and RCC3779 not shown in panel a?

Unfortunately, both cultures were lost to a power outage before the PAM measurements were conducted. We will update the manuscript to clarify this.

---

## Author Response (AR2)

We want to thank the anonymous reviewer for their useful final feedback which helped improve our manuscript's clarity and correctness.

Technical correction from referee:
line 99: strain

*Cell size was measured for each strain in both nutrient replete and nutrient depleted cultures.*

line 107: lower case C

*For replete experiments, growth rates were estimated using change in cell abundance over time as estimated using a fluorescence plate (BMG Labtech) with chlorophyll fluorescence excitation at 420 nm and at 680 nm - which were calibrated against manual Sedgewick Rafter Counter (SRC) counts.*

line 143: I think the terms used should be "nitrogen-replete" and "nitrogen-depleted"?

*Photosynthetic efficiency (Fv/Fm) and electron transport rates (ETR) (Maxwell et al., 2000) were measured using Pulse-Amplitude Modulation (PAM) for both nitrogen-replete and nitrogen-depleted cultures, which was conducted using a Walz WATER-PAM.*

line 147: "plant" might want to be changed to "algal"

*However, in practice, Fv/Fm is generally interpreted as a proxy of algal health.*

line 150 : "of" to "in"

*ETR is also a measure of photosynthetic yield but is measured in light-adapted cells and is a direct measurement of the quantum yield of photochemical energy conversion.*

line 158: micron symbol

*For the nitrogen replete experiment, cultures were grown under 220 µM NO3 and Fv/Fm and ETR were measured during the150 exponential growth phase.*

line 163: not sure what you mean by "lost"

We replaced "lost" with "died"

*Measurements were taken for only PLY182g, RCC6535, RCC3777 and RCC1203. RCC1200 and RCC3779 died before the Fv/FM and ETR measurements were conducted.*

line 344: nitrogen-replete

*Under nitrogen-replete conditions (Fig. 6a), the genome content accounts for 5-15% (10.7 ± 5.8 %) of the nitrogen budget.*

line 455: "other" to "the presence of other" ?

*Our study shows that the genome content of HET cells is twice that of HOL cells, but that, nonetheless, the nitrogen quotas of both phases are similar when nitrogen deplete, which suggests that the presence of other trade-offs not considered here might be essential in sustaining similar minimum nitrogen quotas for HET and HOL cells.*

line 489: nitrogen-replete

*We found that genome content plays a minor role in the nitrogen budget of nitrogen-replete cells, but that the absolute DNA content is higher for HET cells.*

---

## Author Response (AR3)

We would like to extend a big thank you to the editor for spotting some final edits. They were implemented as suggested.